# GTP Binding Protein Gtr1 Cooperating with ASF1 Regulates Asexual Development in *Stemphylium eturmiunum*

**DOI:** 10.3390/ijms23158355

**Published:** 2022-07-28

**Authors:** Shi Wang, Chunyan Song, Lili Zhao, Wenmeng Xu, Zhuang Li, Xiaoyong Liu, Xiuguo Zhang

**Affiliations:** 1Shandong Provincial Key Laboratory for Biology of Vegetable Diseases and Insect Pests, College of Plant Protection, Shandong Agricultural University, Taian 271018, China; wangssdau@126.com (S.W.); a13365487446@163.com (C.S.); zll10062021@163.com (L.Z.); xwm18315476678@126.com (W.X.); liz552@126.com (Z.L.); 2College of Life Sciences, Shandong Normal University, Jinan 250014, China; 622001@sdnu.edu.cn

**Keywords:** GTP binding protein Gtr1, interaction, ASF1, asexual development, *Stemphylium eturmiunum*

## Abstract

The Gtr1 protein was a member of the RagA subfamily of the Ras-like small GTPase superfamily and involved in phosphate acquisition, ribosome biogenesis and epigenetic control of gene expression in yeast. However, Gtr1 regulation sexual or asexual development in filamentous fungi is barely accepted. In the study, SeGtr1, identified from *Stemphylium eturmiunum*, could manipulate mycelial growth, nuclear distribution of mycelium and the morphology of conidia in *Segtr1* silenced strains compared with its overexpression transformants, while the sexual activity of *Segtr1* silenced strains were unchanged. SeASF1, a H3/H4 chaperone, participated in nucleosome assembly/disassembly, DNA replication and transcriptional regulation. Our experiments showed that deletion *Seasf1* mutants produced the hyphal fusion and abnormal conidia. Notably, we characterized that *Segtr1* was down-regulated in Se∆*asf1* mutants and *Seasf1* was also down-regulated in Si*Segtr1* strains. We further confirmed that SeGtr1 interacted with SeASF1 or SeH4 in vivo and vitro, respectively. Thus, SeGtr1 can cooperate with SeASF1 to modulate asexual development in *Stemphylium eturmiunum*.

## 1. Introduction

The fungal hyphae are in a vegetative growth state under normal conditions. The hyphae will form various propagules and then enter the reproduction stage under a certain period. The propagation of filamentous ascomycetes mainly includes asexual reproduction (formation of asexual spores) and sexual reproduction. The main method of asexual reproduction of filamentous fungi is to produce multicellular structures called conidiophores, each bearing asexual spores called conidia [1]. Asexual spores are generated when the surroundings are favorable for growth and development. However, when nutrient availability become scarce, the sexual reproduction occurs and fruit bodies are produced to resist adverse environment [2,3]. Although it is more complicated, it has been found that the evidence of sexual reproduction is confirmed in some eukaryotic groups [4].

Previous studies have shown that fungi are a group historically considered to present a high proportion of asexual species [5]. *Stemphylium*, a genus of ascomycetes in Pleosporales (Dothideomycetes), is known to reproduce asexually [6]. *Stemphylium* is closely related to *Alternaria* and *Ulocladium* [7,8]. The conidia of *Stemphylium* forming on proliferating conidiophores in which produced apically swollen conidiogenous cells, which is principal distinguished it from two closely related genera [6,9]. In addition, the evolution, differentiation and asexual development of filamentous fungi are regulated by several genes, such as velvet family [10,11] and phytochromes [12]. However, another example of the GTP binding proteins with a role in biological development is the Gtr1. The Gtr1 protein is a member of the RagA subfamily of the Ras-like small GTPase superfamily [13]. Moreover, the Gtr1, a multifunctional GTP-binding protein, is involved in phosphate acquisition [14], ribosome biogenesis [13] and epigenetic control of gene expression [15] in *Saccharomyces cerevisiae.* Notably, Gtr1 may be involved in stamen development via active GA supply in *Arabidopsis* [16]. However, Gtr1 regulation sexual or asexual development in filamentous fungi is barely accepted.

In recent years, several genes with potential functions in chromatin modification were found to be involved in asexual and sexual development [17,18,19]. One such factor is the histone chaperone ASF1 that was first identified in *Saccharomyces cerevisiae* [20]. ASF1, a H3-H4 chaperone, is highly conserved from yeast to mammals and involved in nucleosome assembly/disassembly [21,22,23], DNA replication, repair and transcriptional regulation [24]. Interestingly, in our experiments, deletion *Seasf1* mutants produced the hyphal fusion and abnormal conidia. To further investigate asexual development in *S. eturmiunum*, we hypothesized that SeGtr1 could be involved in SeASF1 regulation asexual development. Here, to address the hypothesis, we analyzed RNA transcript levels of the Se∆*asf1* mutant and wild type strain at various stages, and found that SeGtr1 interacted with SeASF1 and SeH4 in *S. eturmiunum*. We confirmed that SeGtr1 could cooperate SeASF1 involved in central regulatory pathway to modulate asexual development.

## 2. Results

### 2.1. SeASF1 Regulates Asexual Development in S. eturmiunum

To understand the biological functions of the SeASF1 during asexual development of *S. eturmiunum*, we obtained two Se∆*asf1* mutants and two complemented transformants. To determine the role of *Seasf1* in hyphal and colonial growth, these four mutants and WT strains were inoculated on CM medium and then photographed after 9 days. Colony growth rates of the two Se∆*asf1* mutants were distinguishable from the WT while two complemented transformants returned to normal growth (Figure 1A). Additionally, two Se∆*asf1* mutants produced the hyphal fusion and abnormal conidia compared to the complemented transformants and WT strains (Figure 1B,D). These results suggest that SeASF1 is involved in asexual development of *S. eturmiunum*.

### 2.2. SeGtr1 Plays a Role in Asexual Development

Gtr1 (SeGtr1) was cloned from *S. eturmiunum*. SeGtr1 contains 335 amino acids with a calculated molecular mass of 37 kDa. To verify the roles of *Segtr1* during the growth and development of *S. eturmiunum*, we obtained two *Segtr1*-silenced transformants (Si*Segtr1*-T63 and Si*Segtr1*-T65) and two overexpression transformants (OE*Segtr1*-T3 and OE*Segtr1*-T8) by *A. tumefaciens* mediated method (Figure 2D). Control was a negative control strain. As a result, two silenced transformants appeared the slow growth rate of colonies related to overexpression transformants or control strains (Figure 2A,C). In addition, the nuclei were anomalously distributed in mycelia of SiSegtr1 strains (Figure 2B). To further observe the roles of Segtr1 during the asexual and sexual development of *S. eturmiunum*, all transformants and WT strains were in the dark condition at 25 °C for 5 weeks on CM medium by inserting double slides. For two silenced strains, the conidiophores turned into bead-like and the conidia grew subglobose, which were significantly different from overexpression transformants and WT strains (Figure 3A). Furthermore, the expression levels of genes involved in the central regulatory pathway, including the *brl*, *aba*, and *wet*, were significantly down-regulated in the two Se∆*asf1* mutants and two *Segtr1*-silenced transformants (Suppplementary Appendix A). However, two silenced strains still produced perithecia that were unanimous with overexpression transformants and WT strains (Figure 3B). These results indicate that SeGtr1 can affect the expression of related genes (*brl*, *aba* and *wet*) in the central regulatory pathway and modulate asexual development.

### 2.3. The Expression Patterns of Seasf1, SeH4, and Segtr1 in Knockout Mutants, Silenced Lines or Overexpression Strains

Due to SeGtr1 can also control the asexual development in *S. eturmiunum*, we verify whether can occur the relation among SeASF1, SeH4 and SeGtr1. Two *SeH4*-silenced strains were Si*SeH4*-T8 and Si*SeH4*-T20. The transcript levels of *Segtr1* and *SeH4* were detected in two Se∆*asf1* mutants and two complemented transformants. The expressions of *Segtr1* and *Seasf1* were measured in two Si*SeH4* lines, and those of *Seasf1* and *SeH4* examined in two Si*Segtr1* lines and two OE*Segtr1* strains. As a result, *Segtr1* and *SeH4* showed down-regulation and up-regulation in two Se∆*asf1* mutants, while *Segtr1* was up-regulation in Se∆*asf1*::*Seasf1* strains, respectively (Figure 4A). At the same time, *Seasf1* displayed up-regulation in two Si*SeH4* lines, but *Segtr1* did not change (Figure 4B). Furthermore, *Seasf1* and *SeH4* showed down-regulation and up-regulation in two Si*Segtr1* lines, respectively. However, *Seasf1* showed up-regulation in two OE*Segtr1* strains (Figure 4C). Summary, SeGtr1 could interact with SeASF1 or SeH4.

### 2.4. SeASF1 Interaction with SeH4, and SeGtr1 Interaction with SeASF1 and SeH4

To test whether can occur the interaction between SeASF1 and SeGtr1, SeH4 and SeGtr1. Y2H revealed that SeASF1 interacted with SeH4, and SeGtr1 interacted with SeASF1 and SeH4 (Figure 5A). On the basis of GST pull-down, SeASF1 was specifically interacted with SeH4 (Figure 5B), while SeGtr1 could interact with SeASF1 and SeH4, respectively (Figure 5C). To further assure those results of the Y2H and pull-down experiments, SeASF1-GFP and SeH4-Flag, SeGtr1-Flag and SeASF1-GFP, and SeGtr1-Flag and SeH4-GFP were expressed in *F. graminearum* protoplasts, respectively, and then the immune complexes were estimated using Co-IP assays (Figure 5D,E). Thus, SeASF1 interacted with SeH4, and SeGtr1 interacted with SeASF1 and SeH4. Altogether, SeGtr1 could cooperate with SeASF1 and SeH4 to modulate asexual development of *S. eturmiunum*.

## 3. Discussion

*Stemphylium* was a dematiaceous hyphomycete that was established with *S. botryosum* as type species [25]. Until now, there were more than 150 *Stemphylium* species had been described [26,27,28,29]. *S. eturmiunum*, a typical species of *Stemphylium* genus, was an important homothallic filamentous fungus, and it could produce both conidia and perithecia. In a previous study, ASF1 could manipulate the sexual reproduction in *Sordaria macrospora* effectively [30]. However, in this study, deletion of *Seasf1* carried out asexual development characters, such as hyphal fusion and abnormal conidia. A possible explanation for the SeASF1 in asexual development was functional redundancy because some genes were down-regulated or up-regulated expression when *Seasf1* was deleted. Therefore, we hypothesized that other proteins might be involved in SeASF1 regulation asexual development.

In summary, through the comparative analysis of transcriptome data (Appendix A), we characterized that the expression of *Segtr1* (No. TR23646-c0_g1) and *Seasf1* (No. KX033515) was down-regulated in Se∆*asf1* and Si*Segtr1* strains, respectively. Meanwhile, we found that SeASF1 or SeGtr1 participated in the central regulatory pathway and regulated the expression of related genes, such as *brl*, *aba* and *wet*. In addition, we verified that SeGtr1 could effectively stimulate asexual activity of *S. eturmiunum* in *Segtr1* silenced strains compared with its overexpression strains. A model of the process is shown in Figure 6. SeASF1 coupled to SeH4 is translocated into the nucleus. In previous studies, ASF1 was reported to regulate DNA replication and damage repair in nucleosome. In this study, we found that SeGtr1 interacted with SeASF1 or SeH4 in vivo and vitro, and SeGtr1 cooperated with SeASF1, which is involved in the central regulatory pathway, to regulate asexual development in *S. eturmiunum*.

## 4. Materials and Methods

### 4.1. Strains and Culture Conditions

*Stemphylium eturmiunum* strain (EGS 29-099) (WT) and all transformants strains were cultured in the dark condition at 25 °C on complete medium (CM), or potato dextrose agar (PDA) medium for mycelial growth assays. *Escherichia coli* DH5α or *Agrobacterium tumefaciens* AGL-1 was incubated in LB (Luria-Bertani) medium at 37 °C or 28 °C, respectively [31].

### 4.2. Plasmid Construction

Deletion strains for *Seasf1* was generated by homologous recombination. The *Seasf1* flanking regions, 1500 bp upstream and 1500 bp downstream, were amplified using primer pairs (Table 1). The resulting PCR products were ligated to the Hygromycin cassette and then transformed into WT. Transformants were screened by PCR with primers (Table 1). In addition, *Seasf1* was cloned into eGPF-pHDT vector for complementation analysis (Table 2). The recombinant plasmid eGFP-pHDT-*Seasf1* was transformed into the *Se*∆*asf1* mutants by *A. tumefaciens* mediated transformation (ATMT) method [32]. Transformants were screened by PCR and western blot.

RNA interference [33] was used for *Segtr1* silencing. The 525 bp cDNA fragments of *Segtr1* was amplified from *S. eturmiunum* with primers (Table 1) and inserted into vector pCIT that flanked to the intron to form silencing construct, respectively [33] (Table 2). The constructed plasmid pCH-*Segtr1* was transformed into *S. eturmiunum* strain by *A. tumefaciens* mediated transformation (ATMT) method [32].

For overexpression analysis, *Segtr1* was cloned from *S. eturmiunum* with primers (Table 1), and then cloned into eGPF-pHDT vector. Subsequently, recombinant plasmid eGFP-pHDT-*Segtr1* was transformed into the Si*Segtr1* lines by ATMT method. Overexpression transformants were screened by qRT-PCR and western blot.

For co-immunoprecipitation (Co-IP) analysis, SeASF1, SeH4, and SeGtr1 were amplified from *S. eturmiunum* with primers (Table 1), and cloned into the pDL2 or pFL7 in yeast (XK125) by recombination approach [34] (Table 2). Recombinant plasmids were then co-transformed into the protoplasts of *Fusarium graminearum* wild-type strain (PH-1). Transformants were also screened by western blot.

### 4.3. RNA Extraction and qRT-PCR

Total RNA was extracted from mycelia of *S. eturmiunum* growing in PDB (Potato Dextrose Broth) cultures using the Fungal RNA Kit (OMEGA Biotechnology, USA). cDNA was generated using the HiScript II QRT SuperMix for qPCR (Vazyme, Nanjing, China). The qRT-PCR was carried out using the 2 × ChamQ SYBR Color qPCR Master Mix (Vazyme, China) and performed on an ABI QuantStudio^TM^ 6 Quantitative Real-Time PCR System (Applied Biosystems). The specific primers of qRT-PCR listed in the Table 1. Changes in the relative expression level of each gene were calculated by the 2^−∆∆CT^ method [35]. Gene expression levels were normalized using the housekeeping gene *actin*. This experiment was repeated at least three times.

### 4.4. Yeast Two-Hybrid

To test whether SeASF1 and SeH4 interact with SeGtr1, Y2H assay was performed according to the Yeast Protocols Handbook (Clontech) using the Y2H Gold yeast reporter strain (Clontech). The *Seasf1* or *SeH4* was amplified and cloned into pGBKT7 (BD). The *Segtr1* or *SeH4* was amplified and cloned into pGADT7 (AD) (Table 2). The primers used are listed in Table 1. Pairwise interaction was tested using AD and BD to transform the yeast strain Y2H gold. The yeast transformants were grown on SD/-Trp/-Leu mediums (TaKaRa Bio) for 3–5 days and then cultured on selection mediums (SD/-Trp/-Leu/-His/-Ade/X-α-gal) to detect the protein-protein interaction. Each experiment was repeated at least three times.

### 4.5. GST Pull-Down

The *Seasf1* was cloned into the pET28a vector after adding a 1×FLAG tag to the 5′-terminal of *Seasf1* to make the Flag-SeASF1-His fusion protein. The *Segtr1* was cloned into the pGEX-6P-1 vector to make the GST-SeGtr1 fusion protein (Table 2). Flag-SeASF1-His, GST-SeGtr1, pET28a or pGEX-6P-1 was expressed in BL21 strain of *E. coli*, and were then affinity purified with a Ni-affinity column (GE) or GST-affinity column (glutathione sepharose^TM^ 4B beads GE Healthcare, Little Chalfont, Buckinghamshire, UK). For glutathione S-transferase (GST) pull-down in vitro, GST-SeGtr1 and Flag-SeASF1-28a were expressed in *E. coli* strain BL21 (DE3). Total proteins of GST-SeGtr1 and Flag-SeASF1-His were then incubated with 4000 μL of glutathione sepharose^TM^ 4B beads at 4 °C for 2 h. The supernatant was removed, and the beads was washed by GST-lysis buffer three times. Finally, the beads were eluted by GST-elution buffer. Pull-down of GST-SeGtr1 with Flag-SeASF1-His was detected using an anti-Flag (Invitrogen, Waltham, MA, USA). Each experiment was repeated at least three times.

### 4.6. Co-IP

*Fusarium graminearum* protoplasts were transfected with the indicated combination plasmids and empty construct. Proteins of *F. graminearum* were extracted in an extraction buffer (50 mM HEPES, 130 mM NaCl, 10% glycerin, pH 7.4) with 25 mM Glycerol phosphate, 1 mM Sodium orthovanada and protease inhibitor (100 mM PMSF). For FLAG IP, protein extracts were incubated with 30 μL of Anti-Flag^®^ M2 Affinity Gel beads (Sigma-Aldrich, St. Louis, MO, USA) at 4 °C for 4 h. The beads were washed five times with by Co-IP washing buffer (50 mM HEPES, 130 mM NaCl, 10% glycerin, pH 7.4). The bound proteins retained on the beads were separated by 12% SDS–PAGE gels and detected using immunoblotting with anti-FLAG (Sigma-Aldrich) or anti-GFP antibody (Invitrogen). Each experiment was repeated at least three times.

### 4.7. Western Blot

Proteins were separated by 12% sodium dodecyl sulfate polyacrylamide gel electrophoresis (SDS-PAGE), and transferred to Immobilon^®^-P PVDF membrane for 1.5 h at 230 mA. The PVDF membranes were blocked with TBST (0.02 M Tris-base, 0.14 M NaCl, 0.1% Tween-20, pH 7.4) with 5% non-fat milk for 1 h at room temperature. Co-immunoprecipitated proteins were analyzed by incubating the membranes with 1:5000 diluted GFP or FLAG antibodies (Sigma-Aldrich) at room temperature for 1–1.5 h. The membranes were washed three times with TBST and then were incubated for 1 h with Goat anti-Mouse-HRP secondary antibody (Thermo Fisher Scientific, Waltham, MA, USA, no. 31430) at 1:7500 dilution. The specific proteins were visualized by the ECL Chemiluminescence Detection Kit (Vazyme) and imaged using a Tanon-5200 System. Each experiment was repeated at least three times.

### 4.8. Microscopy

To observe the morphology of conidia and conidiophores, all the transformants and WT strains were grown in the dark condition at 25 °C for 4 weeks on PDA medium by inserting double slides. Microscopic examination of nuclear distribution in mycelia, the transformants and WT strains were stained using 4,6-diamidino-2-phenylindole (DAPI). To image the sexual structures including perithecia and asci, all these test strains were cultured on CM medium at 25 °C for 6 weeks in dark condition. Perithecia were sectioned by using a double-edged blade in a dissecting microscope (Olympus, SZX10). The asci, conidia and conidiophores were all captured with 20× or 40× objectives of Olympus microscope (Olympus BX53, Tokyo, Japan) using differential interference contrast (DIC) and fluorescence illumination. Microscopic characters of asexual structures were further determined by measurements of 50 mature conidia and 50 conidiophores. The experiment was repeated at least three times.

### 4.9. Statistical Analysis

All determinations were carried out in triplicate and the results are expressed as mean ± standard deviation (SD). The data were subjected to one-way analysis of variance (ANOVA). Statistically significant differences were determined by two-tailed Student’s *t* test: * *p* < 0.05, ** *p* < 0.01.

## Figures and Tables

**Figure 1 ijms-23-08355-f001:**
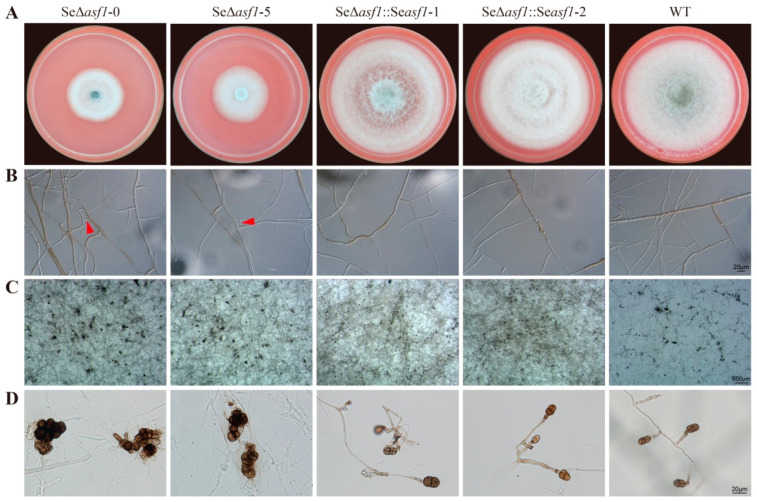
*Seasf1* regulates asexual developmental characterization in *S. eturmiunum.* (**A**) Growth of two Se∆*asf1* mutants, two Se∆*asf1*::*Seasf1* transformants, and WT strains on CM medium. The cultures were photographed after 9 days of incubation. (**B**) Characterizations of hyphal fusion in two Se∆*asf1* mutants, two Se∆*asf1*::*Seasf1* transformants, and WT strains. The images were photographed after growing on PDA medium for 8 days. The fusions in the hyphae were marked with red arrows. (**C**,**D**) Conidia morphology of four mutants and WT strains were cultured on CM medium for 4 weeks. Bar = 20 μm, 500 μm.

**Figure 2 ijms-23-08355-f002:**
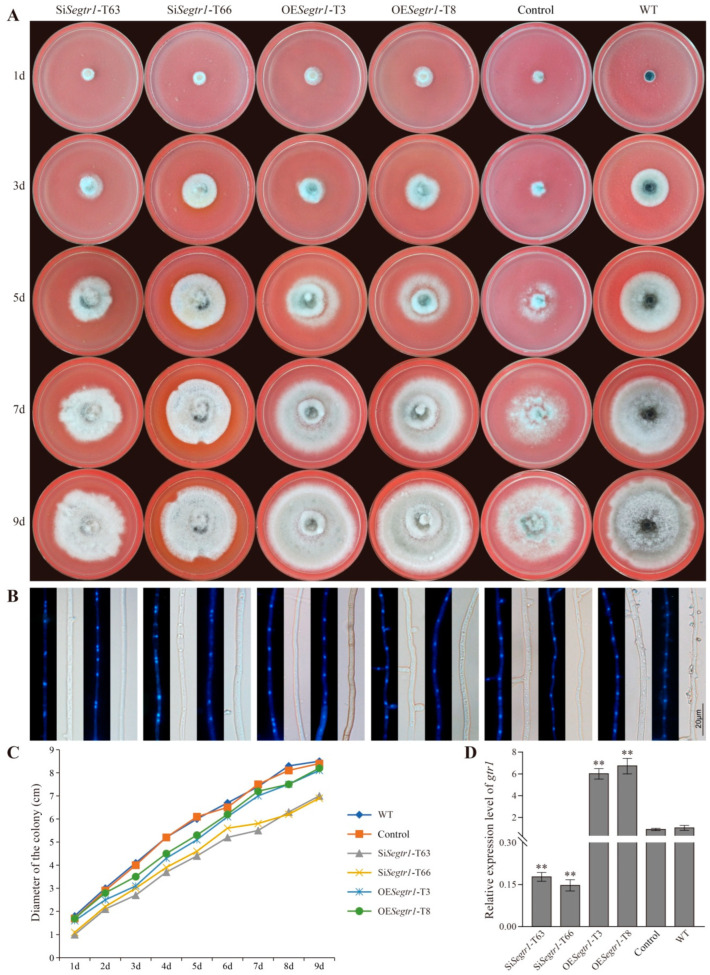
The colonial phenotypes and nuclear distribution of *Segtr1* silenced transformants. (**A**) Colonial growth of two *Segtr1* silenced transformants (Si*Segtr1*-T63 and Si*Segtr1*-T66) and two *Segtr1* overexpression transformants (OE*Segtr1*-T3 and OE*Segtr1*-T8) was observed on PDA medium. WT was *S. eturmiunum* strain and Control was a negative control strain. The cultures were photographed after 1 day, 3 days, 5 days, 7 days and 9 days. (**B**) The mycelium of these four transformants were grown on PDA medium for 6 days and examined by DIC and fluorescence microscopy. The nuclei of the mycelia were discovered under the fluorescence microscopy after staining by DAPI. Bar = 20 μm. (**C**) Colony diameters were measured in each independent biological experiment at 1–9 days of growth on PDA medium. Rates of colonial growth were calculated for each treatment. (**D**) qRT-PCR was used to measure the expression levels of *Segtr1* in silenced transformants, overexpression transformants, Control and WT. The degree of WT was assigned to value 1.0. *Actin* gene of *S. eturmiunum* was used as endogenous control. The bars indicated statistically significant differences (ANOVA; ** *p* < 0.01). Each experiment was repeated at least three times.

**Figure 3 ijms-23-08355-f003:**
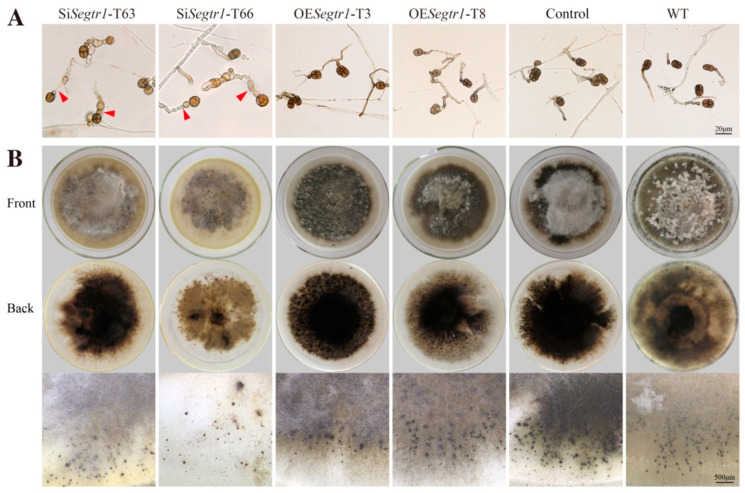
SeGtr1 plays a role in asexual development, but the sexual activity was unchanged. (**A**) For the microscopic investigation of conidiophores and conidia development, two silenced transformants (Si*Segtr1*-T63 and Si*Segtr1*-T66), two overexpression transformants (OE*Segtr1*-T3 and OE*Segtr1*-T8), Control and WT strains were grown on CM medium for 35 days, respectively. Control was a negative control strain. Red arrowheads indicated abnormal conidiophores. (**B**) To further observe the role of *Segtr1* during the sexual development of *S. eturmiunum*, all transformants and WT strains were cultured on PDA medium for inducing perithecia production. At 35 days, all transformant strains produced abundant perithecia. Bar = 20 μm and 500 μm.

**Figure 4 ijms-23-08355-f004:**
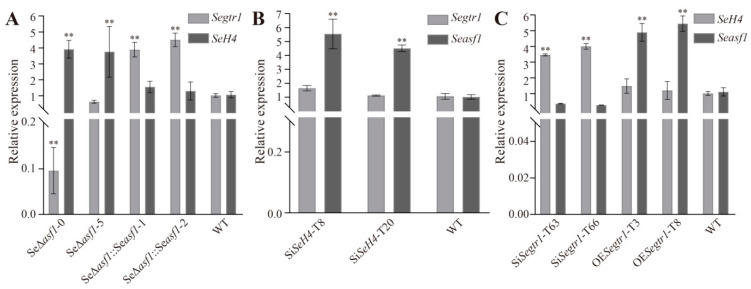
The expression patterns of *Seasf1*, *SeH4*, and *Segtr1* in knockout mutants, silenced lines or overexpression strains. (**A**) The expression levels of *Segtr1* and *SeH4* in two Se∆*asf1* mutants and two complemented transformants were measured by qRT-PCR. (**B**) The expression levels of *Segtr1* and *Seasf1* in two Si*SeH4* lines were measured by qRT-PCR. (**C**) The expression levels of *Seasf1* and *SeH4* in two Si*Segtr1* lines and two OE*Segtr1* lines were measured by qRT-PCR. The degree of WT was assigned to value 1.0. Two *Seasf1* deleted mutants were Se∆*asf1*-0 and Se∆*asf1*-5, two complemented transformants were Se∆*asf1*::*Seasf1*-1 and Se∆*asf1*::*Seasf1*-2. Two *SeH4*-silenced lines were Si*SeH4*-T8 and Si*SeH4*-T20. Two *Segtr1*-silenced lines were Si*Segtr1*-T63 and Si*Segtr1*-T66, two OE*Segtr1* lines were OE*Segtr1*-T3 and OE*Segtr1*-T8. The *Actin* in *S. eturmiunum* was used as endogenous control. The bars indicated statistically significant differences (ANOVA; ** *p* < 0.01).

**Figure 5 ijms-23-08355-f005:**
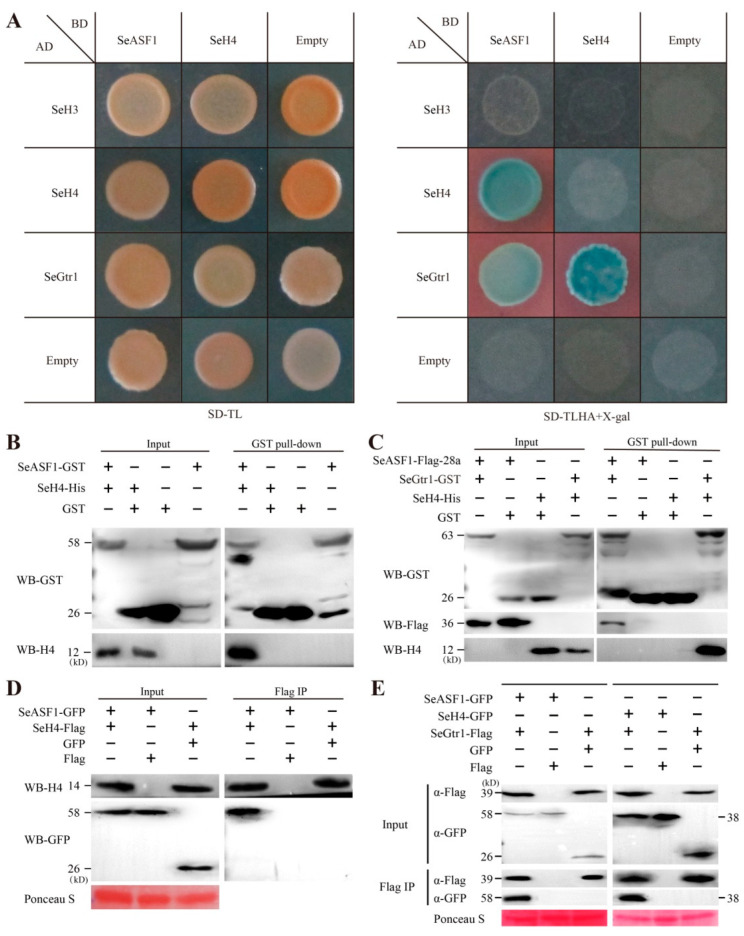
SeASF1 interaction with SeH4, and SeGtr1 interaction with SeASF1 or SeH4. (**A**) SeASF1 interacted with SeH4, and SeGtr1 interaction with SeASF1 or SeH4 using Y2H. SeASF1 or SeH4 was cloned into plasmid pGBKT7 (BD). SeH4 or SeGtr1 was cloned into plasmid pGADT7 (AD). Yeast transformants were first grown on SD/-Trp/-Leu, and selected on SD/-Trp/-Leu/-His/-Ade/X-α-gal. A positive interaction results in the activation of the *lacZ* reporter, which turned the blue in the presence of X-α-galactosidase. The images were photographed at 3 days after incubation. (**B**) SeASF1 was cloned into plasmid pGEX-6P-1. SeH4 was cloned into plasmid pET28a. SeASF1-GST was expressed in *E. coli* and incubated with SeH4-His, purified (pull-down) by glutathione sepharose beads. Recombinant GST was control. SeH4-His was pulled down by SeASF1-GST. (**C**) SeGtr1 was cloned into plasmid pGEX-6P-1. Flag-SeASF1 or SeH4 was cloned into plasmid pET28a. SeH4-His and Flag-SeASF1-His were both retained by SeGtr1-GST. (**D**) SeASF1 was cloned into plasmid pDL2, SeH4 was cloned into plasmid pFL7. Total proteins were extracted from *F. graminearum* protoplasts expressing SeASF1-GFP and SeH4-Flag. Recombinant GFP or Flag was control. The immune complexes were immunoprecipitated with α-Flag antibody (α-Flag IP). Coprecipitation of SeH4-Flag was detected by immunoblotting. (**E**) SeH4 was cloned into plasmid pDL2. SeGtr1 was cloned into plasmid pFL7. Total proteins were extracted from *F. graminearum* protoplasts expressing SeASF1-GFP, SeH4-GFP, and SeGtr1-Flag. Coprecipitation of SeGtr1-Flag was detected by immunoblotting. Membranes were stained with Ponceau S to confirm equal loading. Protein sizes are indicated in kDa. Each experiment was repeated at least three times.

**Figure 6 ijms-23-08355-f006:**
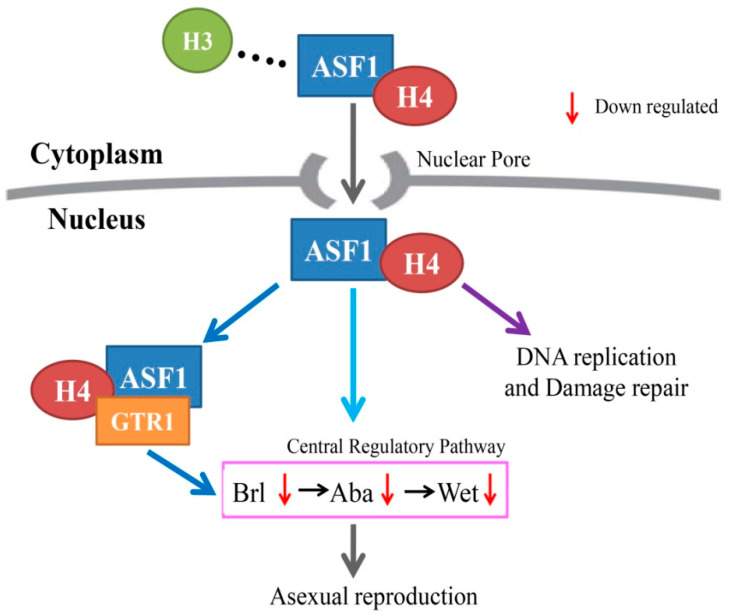
A model for ASF1 binding H4 (ASF1-H4) to interact with GTR1 and then to mediate asexual reproduction in *Stemphylium eturmiunum*. ASF1 interacts with H4 and then they are translocated into nuclei through the nuclear pore. The dimer of ASF1-H4 regulates DNA replication and damage repair in previous studies. Herein, ASF1-H4 combines with GTR1 to constitute a trimeric complex which in turn is involved in the central regulatory pathway to modulate asexual reproduction.

**Table 1 ijms-23-08355-t001:** Primers used in this study.

Primer	Sequence(5′-3′)	Application
*Seasf1-F*	ATGTCTGTCGTTTCGCTTC	Amplify *Se**asf1* sequence
*Seasf1-R*	CTAGTGAACCATGACATCTGC	
*Seasf1-5f*	CCGCTCGAGCAATCCAGGGGCGATAAAG	Amplify *Se**asf1* upstream sequence
*Seasf1-5r*	GGAAGATCTGCTTGGCGGGGTAGATAGAG	
*Seasf1-3f*	CGCGGATCCGCCGCCGTCTGTTAGTCTT	Amplify *Seasf1* downstream sequence
*Seasf1-3r*	CTAGTCTAGACAGAGGAGATGCTTGCTTGTC	
*Seasf1-f*	TCTGTCGTTTCGCTTCTCG	For identification of *Seasf1* deletion strains
*Seasf1-r*	TGAACCATGACATCTGCGC	
*Seasf1-3D*	CCTCGTCGTCTTCCTGATCACT	For identification of *Hph* in deletion transformants
*Hph-3D*	GAGATTCTTCGCCCTCCGAG	
*Hph-5D*	AATTTCGATGATGCAGCTTGGG	
*Seasf1-5D*	CAAGACTGCTTCCTTCTCATCGT	
*Hph-F*	CGACAGCGTCTCCGACCTGA	For identification of *Hph* For identification of
*Hph-R*	CGCCCAAGCTGCATCATCGAA	
*Seasf1-*PHDT*-F*	CCGCTCGAGATGTCTGTCGTTTCGCTTCTCG	*Seasf1*complementation expression
*Seasf1-*PHDT*-R*	GCTCTAGACTAGTGAACCATGACATCTGCGC	
PHDT-*F*	GATCACATGGTCCTGCTG	Vector construction of sequencing primer
PHDT-*R*	CACCAACGATCTTATATCCAG	
*Segtr1-*pCIT-*F*(*BamHI/ClaI*)	CGCGGATCCATCGATGTAGTGCACCTGTTGA ATGAGAGC	Primer for *Segtr1* silence
*Segtr1-*pCIT-*R*(*PstI/EcoRV)*	AACTGCAG GATATCGCCCAGCC TACGTAC ATCCTT	
*Seasf1-QRT-F*	ACAACGAGTACACAGATGAGG	QRT-PCR for *Seasf1*
*Seasf1-QRT-R*	TCGTCAGAGTCCCATTTGATAG	
*SeH4-QRT-F*	AGCCATGATCTACGAGGAAAC	QRT-PCR for *SeH4*
*SeH4-QRT-R*	CGAGAGAAGTGACAGTCTTACG	
*Segtr1-QRT-F*	AAGCTAGCGAGGGTTTCAAG	QRT-PCR for *Segtr1*
*Segtr1-QRT-R*	GGCGTTTGGTATCAGGTAGTAG	
*Seactin-F*	GTCGATTGGAGAAGGAGCTAAA	QRT-PCR for *Seactin*
*Seactin-R*	GTTCTCCTTGTCGGCCATAAT	
*Seasf1-*BD*-F*(*NdeI*)	GGAATTCCATATGATGTCTGTCGTTTCGCTTCTCG	*Seasf1* recombined into pGBKT7
*Seasf1-*BD-*R*(*BamHI*)	CGCGGATCCCTAGTGAACCATGACATCTGCG	
*SeH4*-BD-*F*(*SmaI*)	TCCCCCGGGATGACTGGTCGCGGTAAAGGT	*SeH4* recombined into pGBKT7
*SeH4*-BD-*R*(*BamHI*)	CGCGGATCCCTAACCACCGAAACCGTAAAGGG	
*SeH4*-AD-*F*(*SmaI*)	TCCCCCGGGATGACTGGTCGCGGTAAAGGT	*SeH4* recombined into pGADT7
*SeH4*-AD-*R*(*BamHI*)	CGGGATCCCTAACCACCGAAACCGTAAAGGG	
*SeH3*-AD-*F*(*SmaI*)	TCCCCCGGGATGCCGCCAAAATCCCCTACCAG	*SeH3* recombined into pGADT7
*SeH3*-AD-*R*(*BamHI*)	CGCGGATCCTCAGACAGGCGCCCCCCAAG	
*Segtr1*-AD-*F*(*SmaI*)	TCCCCCGGGATGAATTCAGTCAAGCGTCAGA	*Segtr1* recombined into pGADT7
*Segtr1*-AD-*R*(*BamHI*)	CGCGGATCCCTACATTCCAGAGCCATGC	
*Seasf1-*pET28a-Flag*-F*(*NdeI*)	GGAATTCCATATGGATTACAAGGACGACG ATGACAAGATGTCTGTCGTTTCGCTTCTC	*Seasf1* recombined into pET28a
*Seasf1-*pET28a*-*Flag*-R*(*HindIII*)	CCCAAGCTTCTAGTGAACCATGACATCTGC	
*Seasf1-*pGEX*-F*(*BamHI*)	CGCGGATCCATGTCTGTCGTTTCGCTTCTCG	*Seasf1* recombined into pGEX
*Seasf1-*pGEX*-R*(*NotI*)	ATAAGAATGCGGCCGCCTAGTGAACCATG ACATCTGCG	
*Segtr1-*pGEX*-F*(*NdeI*)	GGAATTCCATATGATGAATTCAGTCAAGCG TCAGA	*Segtr1* recombined into pGEX
*Segtr1-*pGEX*-R*(*NotI*)	ATAAGAATGCGGCCGCCTACATTCCAGAG CCATGC	
*Seasf1-*pDL2*-*GFP-*F*	CAGATCTTGGCTTTCGTAGGAACCCAATCTTCA ATGTCTGTCGTTTCGCTTCTCG	*Seasf1* recombined into pDL2
*Seasf1-*pDL2*-*GFP-*R*	CACCACCCCGGTGAACAGCTCCTCGCCCTTGCT CACGTGAACCATGACATCTGCGC	
*SeH4-*pDL2*-*GFP-*F*	CAGATCTTGGCTTTCGTAGGAACCCAATCTTCA ATGACTGGTCGCGGTAAAGGT	*SeH4* recombined into pDL2
*SeH4-*pDL2*-*GFP-*R*	CACCACCCCGGTGAACAGCTCCTCGCCCTTGCT CACACCACCGAAACCGTAAAGGG	
*SeH4-*pFL7*-*Flag-*F*	CAGATCTTGGCTTTCGTAGGAACCCAATCTTCA ATGACTGGTCGCGGTAAAGGT	*SeH4* recombined into pFL7
*SeH4-*pFL7-Flag-*R*	CTTTATAATCACCGTCATGGTCTTTGTAGT CACCACCGAAACCGTAAAGGG	
*Segtr1-*pFL7*-*Flag-*F*	CAGATCTTGGCTTTCGTAGGAACCCAATCTTCA ATGAATTCAGTCAAGCGTCAGA	*Segtr1* recombined into pFL7
*Segtr1-*pFL7-Flag-*R*	CTTTATAATCACCGTCATGGTCTTTGTAGT CATTCCAGAGCCATGC	
*Segtr1-*PHDT*-F*(*SnaBI*)	CCTACGTA ATGAATTCAGTCAAGCGTCAGA	Primer for *Segtr1* overexpression
*Segtr1-*PHDT*-R*(*XbaI*)	GCTCTAGACATTCCAGAGCCATGC	

**Table 2 ijms-23-08355-t002:** Plasmids used in this study.

Name	Origin of Target Genes	Construction Path	Purpose
*Seasf1*-pXEH	*S. eturmiunum*	*Seasf1* recombined into pXEH	Knockout *asf1*
*Seasf1-*pHDT	*S. eturmiunum*	*Seasf1* recombined into pHDT	*asf1*complementation expression
*Segtr1-*pCIT	*S. eturmiunum*	*Segtr1* recombined into pCIT	*Segtr1* silence
*Seasf1-*pGBKT7	*S. eturmiunum*	*Seasf1* recombined into pGBKT7	Yeast two-hybrid assays
*SeH4-*pGBKT7	*S. eturmiunum*	*SeH4* recombined into pGBKT7	Yeast two-hybrid assays
*Segtr1-*pGADT7	*S. eturmiunum*	*Segtr1* recombined into pGADT7	Yeast two-hybrid assays
*SeH4*-pGADT7	*S. eturmiunum*	*SeH4* recombined into pGADT7	Yeast two-hybrid assays
*SeH3*-pGADT7	*S. eturmiunum*	*SeH3* recombined into pGADT7	Yeast two-hybrid assays
*Seasf1-*pET28a	*S. eturmiunum*	*Seasf1* recombined into pET28a	Pull-down assays
*Segtr1-*pGEX-6P-1	*S. eturmiunum*	*Segtr1* recombined into pGEX-6P-1	Pull-down assays
*Seasf1*-pDL2	*S. eturmiunum*	*Seasf1* recombined into pDL2	CO-IP assays
*SeH4*-pDL2	*S. eturmiunum*	*SeH4* recombined into pDL2	CO-IP assays
*Segtr1-*pFL7	*S. eturmiunum*	*Segtr1* recombined into pFL7	CO-IP assays
*SeH4-*pFL7	*S. eturmiunum*	*SeH4* recombined into pFL7	CO-IP assays
*Segtr1-*pHDT	*S. eturmiunum*	*Segtr1* recombined into pHDT	*Segtr1* over expression

## Data Availability

The data supporting our study results are included in the article.

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
