# Peer review of "GTP Binding Protein Gtr1 Cooperating with ASF1 Regulates Asexual Development in Stemphylium eturmiunum"

_ijms, 2022, doi:10.3390/ijms23158355_

Round 1
Reviewer 1 Report
The authors showed that the Gtr1 modulate asexual development of Stemphylium eturmiunum in cooperation with Asf1. The results shown are new to the biology of S. eturmiunum, but the explanation and description on the findings (results) should be more clear and persuasive. Please check the points given below.
1. To support the results from the asf1-deletion (hyphal fusion & abnormal conidiation), expression of orthologous genes for central regulatory pathway (brl, aba, wet) and of other genes for cell wall-modifying enzymes should be examined.
2. Again, the expression of orthologous genes for central regulatory pathway (brl, aba, wet) and of other genes for cell wall-modifying enzymes should be examined with the gtr1-deletion strains.
3. If possible, please show the expression of sexual gene (magl) for monoacylglycerol lipase in the gtr1-deletion strains.
4. Please show the quantitative data on the perithecia production (numbers/mm2) with the gtr1-deletion. (in relation with the data in Fig 3)
5. In Discussion section, please give a diagram for the interaction of genes (proteins) such as asf1, grt1, and h4 .
6. Please put the gene (or ortholg) name in Table 3, because it is impossible to see that the gtr1 is down regulated by asf1-deletion (in page 8, line 191).
Minor points
1. Title: regulate à regulates
2. What is CK strain (in page 3, line 87)?
3. Please give explanation on the use of Fusarium graminearum for pull-down assay (in page 6, line 155)..
Author Response
Dear sir/madam,
Thank you for your letter and comments concerning our manuscript “GTP binding protein Gtr1 cooperating with ASF1 regulates asexual development in Stemphylium eturmiunum”.Your comments are highly insightful and help us greatly improve the quality of our manuscript. We hope that the revisions and our responses as in the postscript would be sufficient to make our manuscript suitable for publication in International Journal of Molecular Sciences.
Sincerely yours,
Xiuguo Zhang

Reviewer 2 Report
ijms-1791569
GTP binding protein Gtr1 cooperating with ASF1 regulate 2 asexual development in Stemphylium eturmiunum
Authors in present paper described the impact of Gtr1 on asexual activity of Stemphylium eturmiunum. In my opinion article in present form is not ready to be published. First of all there is lack of main goal of research and the reader do not know why those experiments are important. Moreover the discusion section should be improved. Without clearly fourmoluated aim and proper discussion is difficult to revew this work. Also, there are so minor proofreading mistaces (different fonts or diffrerent seize of fonts). Some of them are pointed bellow.
Figure 1/3 - use the same font in the description Put the information about differences between strains. Increase the size of the arrows, scal bars, etc.
L68 - Please explain what CM stand for. Also put the desrirption of others abbreviations, wwhen you used it foor the first time.
Figure 2D/4C- put the inforrmation what data did you comapred.
L213 - please put the full description of PCR and Western Blot analysis.
Author Response

(The authors gave the same response as above.)

Round 2
Reviewer 1 Report
The authors gave proper responses to most of the points raised before. However, there are still some points to be checked for improvement of the manuscript. Please check the points given below.
1. In order to postulate that “SeGtr1 has no effect on sexual development in S. eturmiunum.” (at the last sentence of page 4). the data (or observation) on the sexual development, such as mal gene expression and/or perithecium production, should be provided (or described in the text). Otherwise delete the description on the sexual development in the end of page 4.
2. At the 1st sentence of 2nd paragraph in the ‘Discussion’, please put the corresponding gene ID No. (TR23646-c0_g1) just after the Segtr1 and Seasf1 (if needed), too.
3. It would be better to give explanation on the CK strain and CoIP with F. graminearum (instead of S. eturmiunum ) in the text. Frankly speaking, descriptions in some parts are not fully-explanatory nor easily understandable.
Author Response
Dear sir/madam,
Thank you for your letter again.
Your comments are highly insightful and help us greatly improve the quality of our manuscript “GTP binding protein Gtr1 cooperating with ASF1 regulates asexual development in Stemphylium eturmiunum”. We hope that the revisions and our responses as in the postscript would be sufficient to make our manuscript suitable for publication in International Journal of Molecular Sciences. Please see the attachment.
Sincerely yours,
Xiuguo Zhang
